# Antibiotic Prescription in the Community-Dwelling Elderly Population in Lombardy, Italy: A Sub-Analysis of the EDU.RE.DRUG Study

**DOI:** 10.3390/antibiotics11101369

**Published:** 2022-10-07

**Authors:** Federica Galimberti, Manuela Casula, Elena Olmastroni, Alberico L Catapano, Elena Tragni

**Affiliations:** 1IRCCS MultiMedica, via Milanese 300, 20099 Sesto San Giovanni, Italy; 2Epidemiology and Preventive Pharmacology Service (SEFAP), Department of Pharmacological and Biomolecular Sciences, University of Milan, via Balzaretti 9, 20133 Milan, Italy

**Keywords:** anti-bacterial agents, antibiotic consumption, inappropriate prescribing, elderly

## Abstract

Inappropriate consumption and over-prescription of antibiotics have been extensively reported. Our aim was to specifically evaluate the antibiotic prescribing patterns and appropriateness among the elderly (≥65 years) from the Lombardy region (Italy) in primary care. Antibiotic consumption (as DID: DDD/1000 inhabitants × day) and prevalence rates in 2018 were assessed, and the prescribing quality was evaluated using ESAC-based indicators and WHO-AWaRe criteria. A multivariate logistic regression analysis was performed to evaluate the association between the probability of receiving an antibiotic prescription and patients’ and physicians’ characteristics. A total of 237,004 antibiotic users were included (mean age ± SD 75.98 ± 7.63; males 42.7%). Antibacterial consumption was equal to 17.2 DID, with values increasing with age in both males and females. The study found that the proportion of patients with at least one antibiotic prescription in 2018 was around 39.1%, with different age-related trends between males and females. Consumption (as DID) of cephalosporines (65–74 years: 1.65; 75–84 years: 2.06; ≥85 years: 2.86) and quinolones (3.88, 4.61, 4.96, respectively) increased with growing age, while consumption of penicillins (6.21, 6.08, 6.04, respectively) and macrolides, lincosamides, and streptogramins (3.25, 2.91, 2.64, respectively) decreased. In 2018, antibiotics considered to have higher toxicity concerns or resistance potential, as reported by WHO-AWaRe tool, were consumed more intensively than those to be used as first choices, independent of age and sex. The probability of receiving an antibiotic prescription was greater in females, in subjects with polypharmacy, in treatment with respiratory drugs, anti-inflammatory agents or glucocorticoids, and with previous hospitalization; but increasing age was less associated with exposition to antibiotics.

## 1. Introduction

Antibiotic resistance (ABR) represents a major public health concern globally. Although it can develop naturally, inappropriate use (misuse and overuse) of antibiotics is the key driver [1]. Antibiotic resistance leads to higher medical costs, prolonged hospital stays, and increased mortality [2]. Monitoring and optimizing antibiotics prescription and use represent one cardinal approach to dealing with this pandemic phenomenon [3,4].

During the recent decades, a number of tools have been developed in order to evaluate the quantity and/or the quality of antibiotic consumption in the community, such as the European Surveillance of Antimicrobial Consumption (ESAC)-based quality indicators for antibiotic use in Europe, which consist of a set of 12 indicators [5]. In this regard, the World Health Organization (WHO) has recently developed a tool aimed at improving the quality of antibiotic prescriptions, and, consequently, to decrease the spread of resistant microorganisms and reduce adverse reactions and costs to the health system: the AWaRe classification. This tool classifies antibiotics into three groups, taking into account the impact of different antibiotics and antibiotic classes on ABR, to emphasize the importance of their appropriate use: the Access group includes antibiotic classes to be used as first and second choices for the empirical treatment of 21 common or severe clinical syndromes; the Watch group includes antibiotic classes that are considered to have higher toxicity concerns or resistance potential, compared with the Access one; the Reserve group includes antibiotics that should be used as last-resort options [6,7].

Despite guidelines and programs by government agencies, professional organizations, and medical societies to enhance the quality of antibacterial prescription and reduce the rate of inappropriate antibiotic utilization, prescribing patterns were only slightly changed, and antibiotic use remains approximately steady, particularly in high-income countries, or even shows an upward trend in low- and middle-income countries [8].

This scenario becomes even more alarming among vulnerable populations, such as elderly people, often prescribed with multiple drugs [9]. In Italy, which is the European country with the highest number of elderly subjects (≥65 years), the percentage of elderly patients receiving an antibiotic in 2019 was around 50%, rising to over 60% in men aged 85 years or older, with a consumption of 31 DDD (defined daily doses)/1000 inhabitants per day, comparable to that of non-steroidal anti-inflammatory drugs or pain medications [10,11].

Since the prevention, diagnosis, and management of infectious diseases in elderly patients poses significant and evolving challenges [12], more definitive evidence on antibiotic use and its appropriateness and quality in the elderly is needed to guide interventions in this critical population. Indeed, although the overuse and misuse of antibiotics is well known, and several initiatives have been put in place to tackle the problem, the goal remains to be achieved [13]. More specific description of antibiotic use in specific subpopulations could provide the information to implement targeted interventions. Therefore, we sought (i) to investigate the pattern of antibiotic prescriptions among elderly subjects of both sexes in the primary care setting by analyzing administrative healthcare databases in a large Italian region in 2018, (ii) to assess the quality of antibiotic prescription, using both ESAC-based indicators and WHO-AWaRe tool, and (iii) to identify specific patient- and prescriber-related factors that can negatively affect the antibiotic prescribing in general practice.

## 2. Materials and Methods

### 2.1. Data Source

Data were obtained from the EDU.RE.DRUG study, which have been described in detail elsewhere [14]. Briefly, the EDU.RE.DRUG data were retrieved from administrative databases (demographic, pharmacy-refill, and hospitalization databases) containing healthcare information of all beneficiaries of the National Health Service (NHS). For this study, data for four local health units (LHUs) (Bergamo, Lecco, Mantova and Monza Brianza) in the Lombardy region were used (Appendix A). In particular, the archive of LHU’s residents assisted from the Italian NHS contains some demographic variables (sex, date of birth, date of death); the pharmacy-refill database contains data about all prescriptions of drugs reimbursed by NHS made by general practitioners (GPs) to patients living in the LHUs and dispensed by the pharmacies of the Italian territory (i.e., delivery date, Anatomical Therapeutic Chemical (ATC) code, number of drug’s packages prescribed for each prescription); the hospital discharge archive records, recording information on the admission date and primary and secondary diagnoses of all hospitalizations at public or private hospitals. Compliance with national and European laws on personal data was guaranteed by LHUs through the generation of unique anonymous codes for each patient and each prescriber, with respect to the privacy of every citizen, which is also useful for database matching.

### 2.2. Study Population

The study population included all community-dwelling elderly patients (≥65 years) of the LHUs involved (n = 606,860—retrieved from ISTAT website—https://demo.istat.it/), corresponding to 26.7% of the elderly inhabitants in the entire Lombardy region. We then selected elderly subjects with at least one prescription of antibacterial for systemic use (ATC group J01, according to the ATC classification system) between 1 January 2018 and 31 December 2018.

### 2.3. Evaluation of Antibiotic Use and Consumption

The use of antibiotic drugs in elderly population was evaluated calculating the prevalence of users, therefore, dividing the numbers of subjects with at least one antibiotic prescription between 1 January 2018 and 31 December 2018 by the total number of subjects aged 65 years or older. Prevalence was stratified by age groups (categorized as 65–74, 75–84, and ≥85 years) and sex.

The measure used for estimating the extent of antibiotic drug consumption was the number of DDD per 1000 inhabitants per day (DID). The DID amount was calculated as follows: active substance (g [total, per year])/(365× DDD), divided by the number of inhabitants/1000. The number of DID was then calculated by age groups and sex.

Since the DDD is the assumed average maintenance dose per day for a drug in its main indication for adults, the number of dispensed packages and the number of posological units, as well as the number of PID (packages per 1000 inhabitants per day) and PUID (posological units per 1000 inhabitants per day), were also used to estimate the antibiotic drug consumption during the considered period.

### 2.4. Evaluation of Quality of Antibiotic Use

The quality of the prescriptions was evaluated using two indicator sets: ESAC-based indicators [5] (Appendix A). We specifically assess each indicator by age groups and sex.WHO-AWaRe classification, 2021 [15] (Appendix A). We measured the overall use of antibiotics (expressed as DID) and the prevalence rates for each of the AWaRe categories. The DID amount was then calculated by therapeutic/pharmacological subgroup (ATC 3rd level) and by chemical substance (ATC 5th level).

### 2.5. Statistical Analysis

Continuous variables were reported as mean and corresponding standard deviation or median and first and third quartile if not normally distributed. Categorical variables were reported as absolute frequencies and percentages.

Multivariable logistic regression analysis was performed to estimate the association between the probability of receiving at least one prescription of antibiotics (ATC code J01) between 1 January 2018 and 31 December 2018 (the first prescription was considered as index date), and selected covariates characterizing subjects included in the analysis and their GPs, evaluated in the six months preceding the index date. Only for this analysis, for each subject receiving an antibiotic prescription (case), we matched an individual not receiving antibiotics in 2018 by sex and age (± 5 years), assigning to this subject the same index date as the respective case and evaluating the same covariates. The dependent variable assumed value 1 if the patient had been exposed to antibiotics under exam, while it assumed value 0 otherwise. Model estimates are presented as odds ratios (OR) and the corresponding 95% confidence interval (95% CI). Since fluoroquinolones were among the most commonly used antibiotics and subject of a great deal of attention by regulatory authorities (special warnings and precautions for use, especially in the elderly, due to possible long-lasting adverse drug reactions) [16], we repeated the logistic regression analysis by specifically investigating the probability of receiving at least one prescription of fluoroquinolones (ATC code J01MA). In details, at patient’s level, from demographic databases, we retrieved birth date and sex of each patient. Using pharmacy-refill databases, the number of medicines dispensed in each quarter preceding the index date was calculated, and the highest number of drugs dispensed in a single quarter was used to define polypharmacy over the 6-month period, then defining ‘polypharmacy’ and ‘excessive polypharmacy’ as between 5 and 9 and 10 or more drug-substances prescribed and dispensed to a patient in the same quarter, respectively [17]. Using pharmacy-refill data, we also assessed whether subjects had received at least one prescription of drugs for obstructive airway diseases (ATC code R03), anti-inflammatory and antirheumatic agents for systemic use (ATC code M01A), or glucocorticoids (ATC code H02AB) in the 3 months before the index date. Using hospitalization databases, we identified patients who were hospitalized for any cause in the 2 months preceding the index date. At GP’s level, from demographic databases, we retrieved birth date, sex, and number of registered patients with each GP to calculate the percentage of elderly patients. Using pharmacy-refill databases, we estimated the annual number (in 2018) of different chemical substance (ATC 5th level) prescribed by each GP (defined as “drug portfolio”), and the number of different antibiotic drugs prescribed by each GP (defined as “antibacterial portfolio”). All analyses were also adjusted for the LHU of residence.

The data were analysed using the statistical package SPSS Statistics, version 27.0 for Windows (SPSS Inc., Chicago, IL, USA) and SAS version 9.4 (SAS Inc., Cary, NC, USA).

## 3. Results

A total of 237,004 community-dwelling patients aged 65 years or older with at least one antibiotic prescription in 2018 were included in the study cohort (mean age ± SD, 75.98 ± 7.63; male sex, 42.7%). The distribution by age groups and sex is reported in Table 1. The percentage of females, as expected, increased with age, from 54.8% in 65–74 years to 66.3% in ≥ 85years group. Overall, a total of 827,005 packages were dispensed, corresponding to 3,813,501 DDD and 5,183,449 posological units, with age and sex trends in line with the distribution of the entire population (Table 1).

The prevalence of antibiotic use in 2018 was around 39.1%, with females aged 65–74 years showing higher values than males (39.8% vs. 35.7%). This trend reversed with increasing age: in the ≥85 years group the prevalence was 44.1% in males and 39.4% in females (Table 2).

Antibacterial consumption was equal to 17.2 DID in the entire cohort, with values increasing with age in both males and females; however, males aged ≥85 years exceeded females (females vs. males: 17.3 DID vs. 21.8 DID). The same trends were observed for PID and PUID measures (Table 2).

Each subject exposed to antibiotic treatment received on average 16.1 DDD, 3.5 packages, and 21.9 posological units in 2018; community-dwelling elderly patients were always prescribed more than younger and, within each age group, male were always prescribed more than females (Appendix A).

When the ESAC-based indicators were analyzed by age and sex, consumption of cephalosporines (ESAC 3) and quinolones (ESAC 5) was observed to increase with growing age, while consumption of penicillins (ESAC 2) and macrolides, lincosamides, and streptogramins (ESAC 4) decreased, although different patterns were noted between sexes. Inter age group variations were always larger in males, compared to females (Table 3). For instance, the consumption of quinolones (ESAC 5), although increasing with age, showed a smaller increase in females (+20.5%) than in males (+53.8%), reaching a delta of 2.4 DID between the two sexes in the ≥85 age group. Conversely, the ESAC 6, 7, 8, and 9 indicators showed very low differences between the two sexes, and only for ESAC 8 and 9 there was there an increasing trend with age (Table 3). In Lombardy, the consumption of J01 broad was about 12 times higher than the consumption of J01 narrow, and this ratio (ESAC 10) increased with age and was always higher in males (Table 3). Women aged 65–74 years showed the highest percentage of the seasonal variation in antibacterial consumption (ESAC 11), which diminished with age, reaching the same extent between sexes in the oldest group. The same pattern was observed for the seasonality of quinolones (ESAC 12) (Table 3).

In Figure 1, consumption and prevalence rates of J01 stratified by WHO-AWaRe classes are shown. In 2018, there were no prescriptions of antibiotics on the Reserve list; however, antibiotics on the Watch list were consumed more intensively than those on the Access list, in every age group and sex (Figure 1, panel a). The prevalence of subjects exposed to Watch drugs increased with age and was always higher in females, except among ≥85 old patients (females vs. males: 31.4% vs. 34.1%), while for Access drugs prevalence decreased with age in females and slightly increased in males (Figure 1, panel b).

Comparing the distribution of DID amount by antibiotic classes at ATC 3rd level, based on WHO-AWaRe criteria, major differences between males and females were observed for fluoroquinolones (Watch list), whose consumption increased with age in both males (from 4.31 DID to 6.63 DID) and females (from 3.48 DID to 4.20 DID), although less intensively, and for penicillins in combination with beta lactamase inhibitors (Access list), whose consumption increased with age in males (from 5.03 DID to 6.03 DID), and slightly decreased in females (from 5.03 DID to 4.87 DID). Macrolides (Watch list) consumption was approximately stable with age in males, while in females it decreased with age (Appendix A).

Of the 53 antibacterial substances (ATC 5th level) prescribed in 2018 (14 in the Access group and 39 in the Watch group), 20 (only 5 in the Access group) accounted for 99% of the total DID consumed, while 5/6 accounted for 75% of the total consumption (the drug utilization 75%, DU75%), including amoxicillin+clavulanic acid (30.5%, the only one from the Access list), two quinolones (levofloxacin 16.6% and ciprofloxacin 7.3%), two macrolides (clarithromycin 9.3% and azithromycin 8.6%), and with amoxicillin being the 6th in the rank (Appendix A, panel a). No substantial differences were observed across the two sexes; however, a different distribution was evident among the chemical substances belonging to the Watch list, with higher consumption of quinolones in males and macrolides in females (Appendix A, panel b and panel c).

Results from the logistic regression model showed that the probability of receiving a J01 prescription was greater in subjects of female sex, with polypharmacy, in recent/concomitant treatment with respiratory drugs (R03), anti-inflammatory agents (M01A) or glucocorticoids (H02AB), and with previous hospitalisation for any cause; older age seemed to protect subjects from being exposed to at least one antibiotic (Figure 2). The demographic characteristics (sex and age) of the general practitioner (GPs) did not influence antibiotic prescribing. However, a higher number of different drugs used by the GP (either among all ATC codes or among J01 ATC codes only) increased the likelihood of antibiotic prescription. Conversely, having a percentage of patients over 65 at higher than 25% led to a reduced probability for patients to receive an antibiotic.

On the other hand, analysing the probability of receiving a fluoroquinolone (J01MA) prescription, a lower risk for females, compared to males, and a higher risk with increasing age was observed. All other covariates showed the same pattern as for J01 class in general (Figure 3).

## 4. Discussion

Antibiotics are among the most widely used pharmacological agents [18,19]. They are not only over prescribed, but are often used inappropriately [20]; this contributes to the problem of antibiotic resistance, which increases morbidity and mortality and causes substantial economic losses [21,22]. Recent estimates have shown that more than 670,000 infections due to ABR occur in European countries every year, and as a result nearly 33,000 people die from these infections, with Italy ranking first (and worst) with almost 11,000 deaths [23]. This practice is even more problematic in elderly populations, with increasing risk of associated adverse consequences, including drug interactions, side effects related to age or disease-related changes in metabolisms, and risks associated with multidrug-resistant organisms and *Clostridium difficile* infections [12]. Hence, evaluating the use of antibiotics in real-life is essential to optimize their use in primary care settings and to curb the ABR phenomenon [24].

Our study, including data from a large subset of the elderly population of the Lombardy region, showed a high rate of antibiotic prescriptions in 2018 (prevalence: 39.1%; consumption: 17.2 DID, corresponding to 3,813,501 DDD, 827,005 packages and 5,183,449 posological units dispensed), with differences in the patterns between males and females and among age groups. Overall, DID consumption was lower than the Italian mean value (range: 20–30 DID) in the elderly in 2018 [25], as well as than the EU/EEA population-weighted mean consumption (18.4 DID) [26]. 

In the same year, 16.1 DID were consumed in Italy and 13.7 DID in Lombardy, thus indicating a higher use of antibiotics in the elderly than the entire population [25]. This can be explained by physiological changes in specific immune-response patterns [27] and to higher susceptibility to infections [28]. High rates of consumption in advanced age may also be associated with a greater propensity of physicians to prescribe these drugs to the elderly because of fears associated with the consequences of infection in frail and multimorbid individuals [29,30], or with greater demand from patients themselves [30,31]. Although our data could not highlight the underlying reasons for prescribing, it should be noted that the high prescribing rates in the elderly population are confirmed in other studies. In the Netherlands, Haeseker and colleagues [32] showed that antibiotic prescriptions raised for all age categories between 2000–2009, with the most prominent increase in very old patients (≥80 years), leading to more adverse drug events, higher consumption of health care services, and advanced spread of antibiotic resistance. Palacios-Cena et al. also reported an increasing consumption of J01 prescriptions in elderly subjects from 2003 to 2014 in Spain [33]. In Denmark, between 2016–2017, an antibiotic consumption of 14.7 DID (65–74 years) and 41.3 DID (≥85 years) was observed in the elderly [34], while in the period of 2006–2015 in Ontario (Canada), the average consumption among almost 3 million elderly patients was 25.1 DID, with an average prevalence of 40.7% [35].

In our study, the prevalence of antibiotic users increased with age in males (65–74 years: 35.7% vs. ≥85 years: 44.1%), while in females it remained relatively stable over time (65–74 years: 39.8% vs. ≥85 years: 39.4%); similarly, the prescription rate per 1000 inhabitants increased markedly in males (+51.1%), while in females the increment was less dramatic (+16.6%). Standardized consumption, expressed in DID, PID, and PUID, augmented with age in both sexes, with a more marked increase in males (+32.2%, +73.0%, +32.0%, respectively) than females (+4.1%, +41.4%, +5.6%, respectively). Accordingly, males received more prescriptions than females starting from the age of 70 in Italy [25], and, in another study carried out in the Lombardy region between 2000 and 2019, Franchi and colleagues showed that ≥80 old males exceeded females in antibiotic prevalence (females vs. males: 42.2% vs. 46.1% in 2000 and 41.8% vs. 42.8% in 2019) [36]. Conversely, a study in the La Rioja region (Spain) detected a different trend in the two sexes as age increased: females showed a higher consumption than males in all age groups, reaching the maximum difference in the subjects aged 85–89 years (females vs. males: 35.0 DID vs. 25.5 DID, approximately), but also highlighted a very different prescription pattern between sexes in terms of type of antibiotics [37]. Indeed, in our multivariable logistic regression analysis, female sex was a risk factor for antibiotic prescription, but it was associated with a reduced probability of receiving a fluoroquinolone prescription, suggesting a greater attention by GPs on the use of fluoroquinolones in the clinical context of cystitis, a typically female disorder involving frequent inappropriate use, especially in Italy [38,39]. On the contrary, older age was associated with lower risk of antibiotic prescription, but it increased the probability of fluoroquinolone prescription. This observation also emerged from ESAC-based indicators evaluation, where consumption of cephalosporines and quinolones increased with growing age. Given the safety concerns associated with the adverse reactions of the central nervous system (CNS) in the elderly population and with the CNS excitatory effects of quinolones [40], this evidence suggests that attention that should be paid to the elderly may not be sufficient.

In both models, polypharmacy and hospitalization increased the likelihood of receiving an antibiotic prescription. This is in line with the results of another study carried out in the Lombardy region, in which subjects being exposed to polypharmacy or being hospitalized one to three times a year or more were more than twice as likely to receive antibiotics [36]. Regarding GP characteristics, despite their main involvement in antibiotic prescribing practice, in the two regression models, these variables did not significantly affect prescribing performance, except for older GP age, which increased the probability of the prescription of fluoroquinolones. This is in line with other studies in the literature [41,42], as evidence suggested that primary care physicians who are mid- to late-career are more likely to prescribe antibiotics inappropriately in elderly subjects [42]. This could be due to a decreased effect of training over time or to the difficulty in changing an established practice [43].

In our study, we also found that, according to the ESAC-based indicators [5], the consumption of broad-spectrum penicillins, cephalosporins, macrolides (except erythromycin), and fluoroquinolones was about 12 times higher than the consumption of narrow-spectrum penicillins, cephalosporins, and erythromycin (ESAC 10), and this ratio increased with age and was always higher in males. This evidence in the elderly population is even more alarming than that from the Italian community in 2018, which reported an average ratio of broad-spectrum to narrow-spectrum antibiotic consumption of 7.5 (higher than the European average ratio of 2.9) [26], especially considering that the excessive use of broad-spectrum antibiotics promotes the emergence and spread of multidrug-resistant bacteria, responsible for healthcare-associated infections, and represents a global healthcare issue [44,45,46].

Using the WHO-AWaRe classification tool, we also observed that, in the four LHUs of Lombardy, two out of three prescriptions concerned antibiotics included in the Watch list, with slightly higher prevalence in females than in males, and with an increase in this proportion over time; hence, in the ≥85 age group, three out of four prescriptions referred to Watch active substances. In the 65–74 age group, a higher number of women were exposed to Watch drugs (males vs. females: 24.7% vs. 29.1%), while in the ≥85 years group there was a higher prevalence in males (males vs. females: 34.1% vs. 31.4%), which was consistent with a higher prevalence of male subjects exposed to Access prescriptions (males vs. females: 19.6% vs. 16.3%). Consumption followed the same pattern with increasing age but showed a higher rate for Watch antibiotics among men in the older age group (males vs. females: 13.5 DID vs. 10.5 DID). In Italy, the use of active substances in the Reserve group is not fully detectable by administrative databases because these drugs are mainly utilised in the hospital setting and, when available in general practice, they are poorly prescribed by GPs. Despite this, a major effort is still needed to endorse the use of Access antibiotics and, in parallel, reduce the use of Watch antibiotics, especially in individuals aged 85 years or more. Indeed, the WHO global indicator states that Access agents should constitute 60% of total antibacterial consumption [15]. In the analysis of data on the population from 29 EU/EEA countries of the European Surveillance of Antibiotic Consumption network (ESAC-Net) and 15 countries of the WHO Regional Office for Europe (WHO Europe) AMC Network, only 17 of the ESAC-Net countries and 3 countries in the WHO Europe AMC Network, respectively, achieved this target in 2018. Italy was among the countries that did not reach the target, having Access drugs accounting for 47.6% of total antibacterial consumption [47].

Many other studies have evaluated the appropriate prescription of antibiotics in the elderly; however, owing to the great differences in methodology and data, direct comparisons among studies in different regions and countries are not feasible. Nevertheless, the epidemiological and clinical relevance of the problem has always emerged. Two studies in the United States using the Veterans Affairs Healthcare System reported that 50% of antibiotic prescriptions in primary care clinics were considered unnecessary [48]. In Canada, the analysis of the antibiotic prescription rate for 23 specific conditions showed that 30.6% and 24.3% of antibiotics were prescribed for conditions that never or rarely justify the use of antibiotics, respectively [49]. Up to 23% of antibiotic prescriptions in United Kingdom primary care settings are considered inappropriate [50].

### Strengths and Limitations

The results of this study should be interpreted in light of some limitations. As it was conducted in the setting of primary care using administrative databases:Prescription and dispensing do not equate to consumption, with possible overestimation that may result from drugs not consumed;It is difficult to estimate the duration of the prescribed therapy, especially with antibiotics;There is a lack of information about: (1) indication for antibiotic use, which would allow a more accurate assessment of the prevalence of their inappropriate use that is estimated to be around 25% in Italy [11]; (2) drugs not covered by the NHS in primary care, so that the exposure to antibiotics could be underestimated; (3) patients’ characteristics, such as socio-cultural variables, education, living arrangement, and income, which could be relevant in evaluating different patterns of antibiotic use; (4) antibiogram results that could allow the assessment of appropriateness of drug prescription and possible association with the different profiles of antibiotic use and the ABR in sexes and age groups.

Nevertheless, administrative databases themselves are an element of strength, as they collect all the reimbursed drugs dispensed to all citizens covered by the NHS. Moreover, administrative data collection, managed at a regional level, is nationally standardized, extremely accurate, and commonly used for drug utilization research [51]. Our population of about 237,000 community-dwelling elderly patients treated with antibiotics in 2018 corresponded to 40% of the elderly subjects in Lombardy, the largest Italian region. The use of these kind of sources, based on extensive and standardised data collection, allows one to obtain robust evidence with little expenditure of time and money.

## 5. Conclusions

In this observational study including the Lombardy elderly population, we showed that in 2018 antibiotic prescription had not yet reached acceptable levels in terms of quantity and quality. Even if a high consumption rate is not always synonymous of “inappropriateness” in terms of effectiveness/safety at the individual patient level, Italian data from general practice have shown an over-use of antibiotics in clinical conditions where they are not required, also supported by evidence of a high use of fluoroquinolones and consumption of antibiotics in the winter months, often characterized by viral influenza. What is also worrying is the limited use of first-line antibiotics (i.e., amoxicillin) in fragile subjects with multimorbidity and polypharmacy, and the higher prescription rate of drugs belonging to the Watch group, compared to the Access group, thus contributing to an increase in ABR phenomenon, which is already significant in Italy.

Yet, there is a need to develop strategies for monitoring antibiotic consumption, as well as to put in place multilevel and multifaceted interventions for the education of healthcare professionals and their patients, in order to ensure the rational use of antibiotics and to suppress the spread of antibiotic resistance. In Italy, some local initiatives have been promoted, as multifaceted public campaigns addressed to both healthcare professionals and the public, and studies evaluating their effectiveness showed promising results [52,53]. However, sustained effort, funding, and outreach activities are warranted to effectively influence antibiotic prescribing and further promote their appropriate use. Our data could contribute to fill the gap in the knowledge in this field, in the attempt to provide the basis of tailored interventions. The limitations discussed above and the global impact of the problem, however, call for further studies that can, on the one hand, provide more insight into the areas and determinants of inappropriate antibiotic use and, on the other hand, assess the effectiveness of improvement measures in the medium to long term.

## Figures and Tables

**Figure 1 antibiotics-11-01369-f001:**
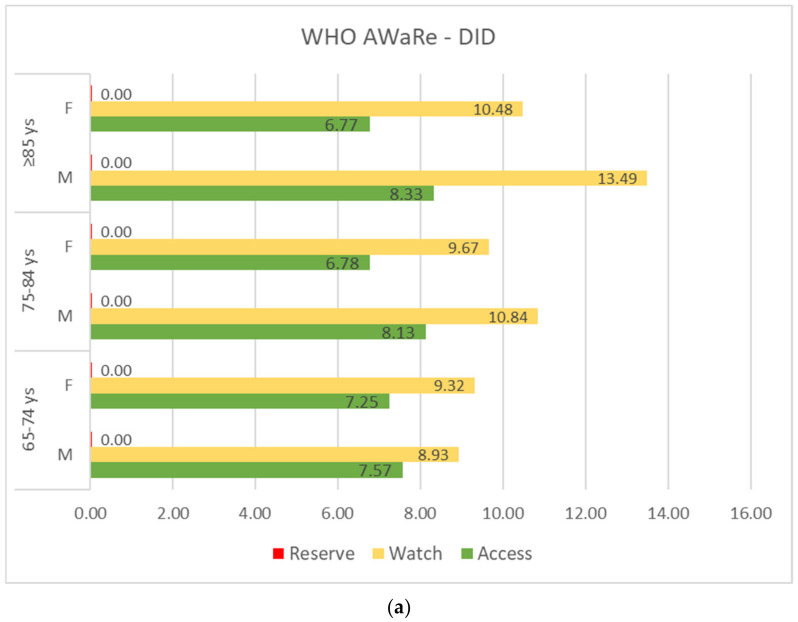
Antibiotic consumption (expressed in DID, panel (**a**) and prevalence rates (%), panel (**b**) according to the WHO-AWaRe (Access, Watch, Reserve) classification system, stratified by sex and age groups.

**Figure 2 antibiotics-11-01369-f002:**
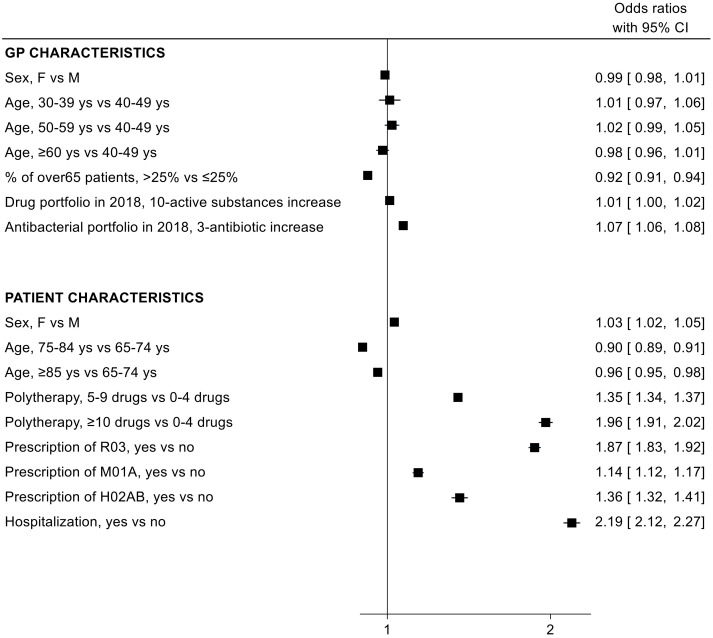
Odds ratio (OR) and 95% confidence intervals (95% CI) for the association between patient/general practitioner (GP) characteristics and the probability of being exposed to at least one antibiotic (J01) prescription.

**Figure 3 antibiotics-11-01369-f003:**
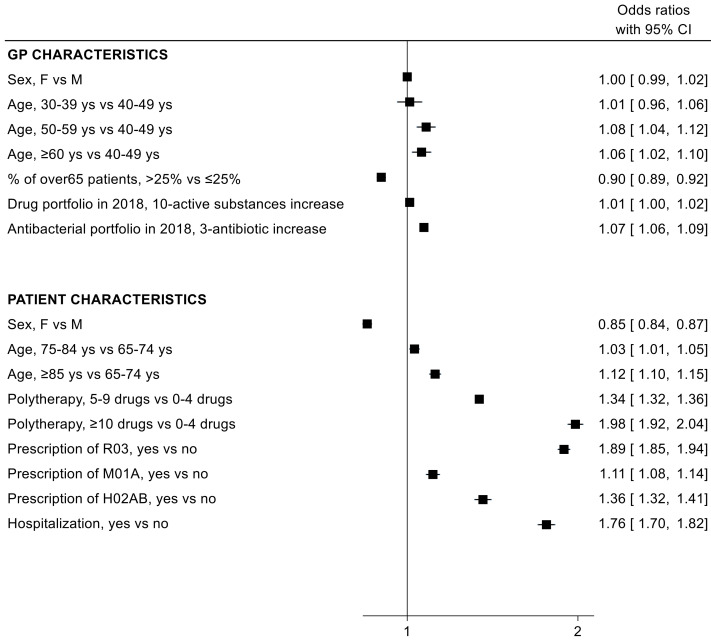
Odds ratio (OR) and 95% confidence intervals (95% CI) for the association between patient/general practitioner (GP) characteristics and the probability of being exposed to at least one fluoroquinolone (J01MA) prescription.

**Table 1 antibiotics-11-01369-t001:** Cohort characteristics, by sex and age groups.

	Overall, N	Males, N (%)		Females, N (%)	
RESIDENT POPULATION					
65–74 ys	299,513	143,277	(47.84)	156,236	(52.16)
75–84 ys	217,733	94,813	(43.55)	122,920	(56.45)
≥85 ys	89,614	27,933	(31.17)	61,681	(68.83)
J01 COHORT					
65–74 ys	113,335	51,159	(45.14)	62,176	(54.86)
75–84 ys	87,048	37,696	(43.30)	49,352	(56.70)
≥85 ys	36,621	12,327	(33.66)	24,294	(66.34)
J01 DEFINED DAILY DOSES (DDD)					
65–74 ys	1808,051	863,257	(47.75)	944,794	(52.25)
75–84 ys	1,394,563	656,515	(47.08)	738,047	(52.92)
≥85 ys	610,887	222,452	(36.41)	388,435	(63.59)
J01 PACKAGES					
65–74 ys	3,57,405	167,959	(46.99)	189,446	(53.01)
75–84 ys	3,07,213	1,41,572	(46.08)	165,641	(53.92)
≥85 ys	162,387	56,652	(34.89)	105,735	(65.11)
J01 POSOLOGICAL UNITS					
65–74 ys	2,450,500	1167,587	(47.65)	1,282,913	(52.35)
75–84 ys	1,897,943	887,727	(46.77)	1,010,216	(53.23)
≥85 ys	835,006	300,378	(35.97)	534,628	(64.03)

**Table 2 antibiotics-11-01369-t002:** Antibiotic prevalence of users and consumption, by sex and age groups.

	Overall	Males	Females
J01 PREVALENCE (%)			
65–74 ys	37.84	35.71	39.80
75–84 ys	39.98	39.76	40.15
≥85 ys	40.87	44.13	39.39
J01 DID (DDD per 1000 inhabitants per day)			
65–74 ys	16.54	16.51	16.57
75–84 ys	17.55	18.97	16.45
≥85 ys	18.68	21.82	17.25
J01 PID (packages per 1000 inhabitants per day)			
65–74 ys	3.27	3.21	3.32
75–84 ys	3.87	4.09	3.69
≥85 ys	4.96	5.56	4.70
J01 PUID (posological units per 1000 inhabitants per day)			
65–74 ys	22.42	22.33	22.50
75–84 ys	23.88	25.65	22.52
≥85 ys	25.53	29.46	23.75

**Table 3 antibiotics-11-01369-t003:** Evaluation of the ESAC-based indicators, by sex and age groups.

	Overall	Males	Females
ESAC 1: Consumption of antibacterials for systemic use (J01) expressed in DID			
65–74 ys	16.54	16.51	16.57
75–84 ys	17.55	18.97	16.45
≥85 ys	18.68	21.82	17.25
ESAC 2: Consumption of penicillins (J01C) expressed in DID			
65–74 ys	6.21	6.08	6.32
75–84 ys	6.08	6.51	5.76
≥85 ys	6.04	6.87	5.67
ESAC 3: Consumption of cephalosporins (J01D) expressed in DID			
65–74 ys	1.65	1.62	1.68
75–84 ys	2.06	2.19	1.95
≥85 ys	2.86	3.16	2.72
ESAC 4: Consumption of macrolides, lincosamides, and streptogramins (J01F) expressed in DID			
65–74 ys	3.25	2.89	3.59
75–84 ys	2.91	2.87	2.94
≥85 ys	2.64	2.99	2.48
ESAC 5: Consumption of quinolones (J01M) expressed in DID			
65–74 ys	3.88	4.31	3.48
75–84 ys	4.61	5.49	3.94
≥85 ys	4.96	6.63	4.20
ESAC 6: Consumption of beta-lactamase sensitive penicillins (J01CE) expressed as percentage of the total consumption of antibacterials for systemic use (J01)			
65–74 ys	0.00082	0.00107	0.00058
75–84 ys	0.00124	0.00206	0.00051
≥85 ys	0.00070	0.00000	0.00109
ESAC 7: Consumption of combination of penicillins, including beta-lactamase inhibitor (J01CR), expressed as percentage of the total consumption of antibacterials for systemic use (J01)			
65–74 ys	30.43	30.46	30.40
75–84 ys	28.77	29.04	28.52
≥85 ys	28.09	27.70	28.31
ESAC 8: Consumption of third- and fourth-generation cephalosporins (J01(DD+DE)) expressed as percentage of the total consumption of antibacterials for systemic use (J01)			
65–74 ys	8.84	8.47	9.18
75–84 ys	10.56	10.36	10.73
≥85 ys	14.22	13.53	14.62
ESAC 9: Consumption of fluoroquinolones (J01MA) expressed as percentage of the total consumption of antibacterials for systemic use (J01)			
65–74 ys	23.40	26.08	20.94
75–84 ys	26.20	28.90	23.79
≥85 ys	26.44	30.37	24.19
ESAC 10: Ratio of consumption of broad-spectrum penicillins, cephalosporins, macrolides (except erythromycin), and fluoroquinolones (J01(CR+DC+DD+(FA–FA01)+MA)) to the consumption of narrow-spectrum penicillins, cephalosporins, and erythromycin (J01(CA+CE+CF+DB+FA01))			
65–74 ys	10.77	11.47	10.19
75–84 ys	12.61	14.18	11.44
≥85 ys	16.90	19.47	15.66
ESAC 11: Seasonal variation (SV) of the total antibiotic consumption (J01)			
65–74 ys	36.62	33.19	40.14
75–84 ys	33.38	32.48	34.33
≥85 ys	34.38	34.37	34.38
ESAC 12: Seasonal variation of quinolone consumption (J01M)			
65–74 ys	29.59	25.65	35.14
75–84 ys	27.09	25.00	30.41
≥85 ys	27.52	25.85	28.26

## Data Availability

Raw data were generated at the local health units. Derived data supporting the findings of this study are available from the corresponding author (MC) on request. Requests to access these data sets should be directed to Manuela Casula, manuela.casula@unimi.it.

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
