# Peer review of "Antibiotic Prescription in the Community-Dwelling Elderly Population in Lombardy, Italy: A Sub-Analysis of the EDU.RE.DRUG Study"

_antibiotics, 2022, doi:10.3390/antibiotics11101369_

Round 1

Reviewer 1 Report

AMR remains a priority, as does ensuring the optimisation of antibiotics/antimicrobials in all care facilities and communities. Hence surveillance/monitoring are key. This study provides important information on the use of antibiotics, using a number of different estimates. 

The study is of interest, and the data may be useful in monitoring the impact of policy/interventions, in this part of Italy, and in comparisons with other countries. It might also act as a way of showing individual clinicians the way in which antibiotics are being used. 

A weakness is that the appropriateness of antibiotics was not fully explored. Whilst a challenge, this is really a key priority, especially as there are increasing numbers of reports from surveillance/individual epidemiological students, only presenting use. It is data on inappropriate use that is a priority. 

The study is of general interest but does not highlight any points of major importance, other than for parts of Italy. 

The paper needs attention:

Abstract

This needs a lot of work and it is confusing as it stands. 

Need to make clear where the subjects are; are they recruited from hospitals or primary care? 

The results shouldn't be an estimate because it is not a sample. It should just report 'the study found that the proportion of patients....

Needs to include more of the actual data from the comparisons. 

Even if Watch or Research classes were used, did they look at whether patients had had failure with first and second line drugs?

The first part of the abstract does not state that they were looking at polypharmacy too. 

I suggest a more structured Abstract is produced, with greater clarity reflecting some of the points above.

Main paper

 Introduction - first and second line confuses AMR with ABR - they need to start with one of these and use throughout, or explain the difference.

It is not clear if the study is about outpatients from hospital clinics, or those in primary care. 

The One Health approach does add anything when mentioned here. The focus of this study is on monitoring/evaluation and Stewardship

Evidence is that the use in high-income countries use is starting to plato, but in low income, it is increasing prolifically, although some of this is positive.

It is important to report the absolute use, and not just the proportion of elderly receiving them.

Stick to one term to describe older people - it goes from older to geriatric to elderly

The objectives need to state polypharmacy if that is to be examined too

Need to give information about the demographics of people in the Lombardy region, and also the number of patient beds, or practice size if based in the community (I'm still a bit confused) in relation to the four local health units, and who these serve.

Not clear how patients were matched across the different databases. What was the single identifier to link them?

Also need to make clear that this is a matched case-control study, at least as indicated line 130

2.3 - need to make clear whether looking at 'ever used', or at least one prescription during a specific time frame (i.e. Jan-Dec 2018), or, the number of prescriptions given that some will have repeated.

The methods need to include polypharmacy - how defined 

259 is repetitive of 254

311 needs supporting references

327 sentence wording needs attention

394 is it worth noting what number of patients are cared for outside of the NHS (e.g. private) - just a thought because I'm not aware of the system in Italy

It would be helpful to include something about the value of Stewardship and how this can be very effective.

Author Response

AMR remains a priority, as does ensuring the optimisation of antibiotics/antimicrobials in all care facilities and communities. Hence surveillance/monitoring are key. This study provides important information on the use of antibiotics, using a number of different estimates.

The study is of interest, and the data may be useful in monitoring the impact of policy/interventions, in this part of Italy, and in comparisons with other countries. It might also act as a way of showing individual clinicians the way in which antibiotics are being used.

A weakness is that the appropriateness of antibiotics was not fully explored. Whilst a challenge, this is really a key priority, especially as there are increasing numbers of reports from surveillance/individual epidemiological students, only presenting use. It is data on inappropriate use that is a priority.

The study is of general interest but does not highlight any points of major importance, other than for parts of Italy.

The paper needs attention:

Abstract

This needs a lot of work and it is confusing as it stands.

Need to make clear where the subjects are; are they recruited from hospitals or primary care?

The results shouldn't be an estimate because it is not a sample. It should just report 'the study found that the proportion of patients....

Needs to include more of the actual data from the comparisons.

Even if Watch or Research classes were used, did they look at whether patients had had failure with first and second line drugs?

The first part of the abstract does not state that they were looking at polypharmacy too.

I suggest a more structured Abstract is produced, with greater clarity reflecting some of the points above.

We understand the point of view of the reviewer. The maximum number of words that can be included in the abstract, however, does not allow us to present the results of our study in more detail. Nevertheless, we modified the abstract in order to address the issues pointed out by the reviewer (now the new version of the abstract contains more words than allowed).

  • We specified that involved subjects were recruited from primary care.
  • Regarding the analysis performed in this study, we did not evaluate failure associated with the prescription of antibiotic belonging to WATCH or RESERVE classes because we performed a cross-sectional analysis about antibiotic use, applying these drug lists which primary aim is the evaluation of the use of different antibiotics and antibiotic classes associated with antibiotic resistance. Although interesting, the assessment of the prescription of inappropriate antibiotics as a second line of treatment after failure of a first line goes beyond the scope of our analysis, and would require additional information, which is not available in our databases, to identify changes in therapy motivated by treatment failure.
  • We did not mention polypharmacy in the first part of the abstract because it was only included as covariate in the logistic model to estimate its effect on the probability to receive an antibiotic/fluoroquinolone prescription.

Main paper

Introduction - first and second line confuses AMR with ABR - they need to start with one of these and use throughout, or explain the difference.

Thank you for your comment. We amended the text, using ABR (antibiotic resistance) throughout the article.

It is not clear if the study is about outpatients from hospital clinics, or those in primary care.

Thank you for your comment. As stated in the introduction, the study sought to assess the pattern of antibiotic prescriptions and the quality of antibiotic prescription among elderly subjects of both sexes in the primary care setting, and to identify specific patient- and prescriber-related factors that can negatively affect the antibiotic prescribing in general practice. To avoid misunderstanding, we changed the text accordingly both in the title and in the text.

The One Health approach does add anything when mentioned here. The focus of this study is on monitoring/evaluation and Stewardship

Thank you for your comment. We deleted the sentence.

Evidence is that the use in high-income countries use is starting to plato, but in low income, it is increasing prolifically, although some of this is positive.

Thank you for your comment. We modified the sentence as follow:

Despite guidelines and programmes by government agencies, professional organizations, and medical societies to enhance the quality of antibacterial prescription and reduce the rate of inappropriate antibiotic utilization, prescribing patterns were only slightly changed, and antibiotic use remains approximately steady, particularly in high-income countries, or even shows an upward trend in low- and middle-income countries [8].

It is important to report the absolute use, and not just the proportion of elderly receiving them.

We agree with this point. We modified the sentence as follow:

In Italy, which is the European country with the highest number of elderly subjects (65 years), the percentage of elderly patients receiving an antibiotic in 2019 was around 50%, rising to over 60% in men aged 85 years or older, with a consumption of 31 DDD (Defined Daily Doses)/1,000 inhabitants per day, comparable to that of non-steroidal anti-inflammatory drugs or pain medications [10,11]”

Stick to one term to describe older people - it goes from older to geriatric to elderly

Thank you for your suggestion. We decided to use “elderly” throughout the entire text.

The objectives need to state polypharmacy if that is to be examined too.

As mentioned above, we prefer not to mention polypharmacy in the aim because it was just a covariate evaluated in the logistic model as patients’ characteristic to estimate its effect on the probability to receive an antibiotic/fluoroquinolone prescription.

Need to give information about the demographics of people in the Lombardy region, and also the number of patient beds, or practice size if based in the community (I'm still a bit confused) in relation to the four local health units, and who these serve.

Thank you for your suggestion. We added Table S1 in the supplementary material with information about resident citizens aged 65 years or older and their general practitioners in Lombardy region and in the four selected local health units.

Not clear how patients were matched across the different databases. What was the single identifier to link them?

Thank you for your comment, that allow us to clarify this point. In all databases, patients are identified by fiscal code. To ensure privacy, the data was anonymised at source, using a unique anonymous code that allows us to link different databases. We added the following sentence in Methods, section 2.1:

Compliance with national and European laws on personal data was guaranteed by LHUs through the generation of unique anonymous codes for each patient and each prescriber, in respect of the privacy of every citizen, also useful for database matching."

Also need to make clear that this is a matched case-control study, at least as indicated line 130

Thank you for pointed this out. We would like to clarify that we did not perform a case-control study; we implemented a match-approach only to perform the logistic regression analysis, to pair each subject receiving an antibiotic prescription with an individual not receiving any antibiotic during 2018, in order to minimize the potential influence of sex and age in comparing individuals with or without prescription of antibiotics. However, in order to avoid misunderstanding, we modified the sentence to:

Only for this analysis, for each subject receiving an antibiotic prescription (case), we matched an individual not receiving antibiotics in 2018 by sex and age (±5 years), assigning to this subject the same index date as the respective case and evaluating the same covariates.

2.3 - need to make clear whether looking at 'ever used', or at least one prescription during a specific time frame (i.e. Jan-Dec 2018), or, the number of prescriptions given that some will have repeated.

We evaluated prevalence of users considering the numbers of subjects with at least one antibiotic prescription in the study period (Jan-Dec 2018). We specified the time frame in the sentence:

The use of antibiotic drugs in elderly population was evaluated calculating the prevalence of users, therefore dividing the numbers of subjects with at least one antibiotic prescription between January 1, 2018 and December 31, 2018 by the total number of subjects aged 65 years or older

The methods need to include polypharmacy - how defined.

Please, note that the way we defined polypharmacy is described at line 148-153 (paragraph 2.5).

259 is repetitive of 254

Thank you. We modified the text accordingly.

311 needs supporting references

Thank you. We added the relevant references (number 38 and 39).

327 sentence wording needs attention

Thank you. We modified the text.

394 is it worth noting what number of patients are cared for outside of the NHS (e.g. private) - just a thought because I'm not aware of the system in Italy

Thank you for this comment. Using administrative database, we cannot estimate the number of subjects who are cared outside the NHS. However, in Italy, the NHS covers all the health services and relative cost, and the proportion of the population using private services is negligible.

It would be helpful to include something about the value of Stewardship and how this can be very effective.

Thank you for your comment. Specifically, in Italy, our National Medicines Agency promotes several initiatives aimed at raising awareness among healthcare professionals and the public on a responsible use of antibiotics. Many initiatives are also organised at local level. Results of some pilot experiences are reported in literature. We added the following sentence in the discussion:

In Italy, some local initiatives have been promoted, as multifaceted public campaigns addressed to both healthcare professionals and the public, and studies evaluating their effectiveness showed promising results [51,52]. However, sustained effort, funding, and outreach activities are warranted to effectively influence antibiotic prescribing and further promote their appropriate use.

Reviewer 2 Report

Journal: Antibiotics (ISSN 2079-6382)

Manuscript ID: antibiotics-1944480

Title: Antibiotic prescription in the elderly outpatient population in Lombardy, Italy: a sub-analysis of the EDU.RE.DRUG study

Please highlight your contributions in introduction. Discuss the novelty and motivation in the last paragraph in the introduction.

What does asterisk sign mean in Figure1.

The caption of Figure 1 MUST be updated. What is reserve?

“In 2018, no prescriptions were found for drugs listed as Reserve”, check this statement.

Add more figures to discuss your results.

The introduction should be supported by recent publications from MDPI such as “Artificial intelligence for forecasting the prevalence of COVID-19 pandemic: an overview”.

Conclusion: What are the advantages and disadvantages of this study.

“Since fluoroquinolones were among the most commonly used antibiotics and subject of a great deal of attention by regulatory authorities”; add a reference.

The inspiration of your work must further be highlighted.

Future work must be included.

Looking and wishes for the revised version.

Author Response

Please highlight your contributions in introduction. Discuss the novelty and motivation in the last paragraph in the introduction.

Thank you for your comment. We added the following sentence to better specify how our analysis could contribute to fill the gap in knowledge in this field, in the attempt to provide the basis of tailored interventions:

Indeed, although the overuse and misuse of antibiotics is well known, and several initiatives have been put in place to tackle the problem, the goal remains to be achieved [13]. More specific description of antibiotic use in specific subpopulations could provide the information to implement targeted interventions.

What does asterisk sign mean in Figure1.

Probably there is a misunderstanding, since there is no asterisk in the Figure 1. However, we are available for any changes, if necessary.

The caption of Figure 1 MUST be updated. What is reserve?

Thank you for your comment. We modified the caption and the Figure 1, accordingly.

“In 2018, no prescriptions were found for drugs listed as Reserve”, check this statement.

Thank you for your comment. We modified the wording, as follow:

In 2018, there were no prescriptions of antibiotics on the Reserve list

Add more figures to discuss your results.

Thank you for your suggestion. We have replaced Table 4 and Table 5 with forest plots (Figure 2 and Figure 3).

The introduction should be supported by recent publications from MDPI such as “Artificial intelligence for forecasting the prevalence of COVID-19 pandemic: an overview”.

We have read the article suggested by the reviewer. Although we found it of great interest, actually it is a review on the applications of different AI approaches used in forecasting the spread of this pandemic. We can not find any mention to antibiotics or drug prescription; moreover, our article did not address issues related to COVID pandemic, since our data refer to 2018. Therefore, we decided not to include the reference.

Conclusion: What are the advantages and disadvantages of this study.

Thank you for pointing this out. We conducted a study using already available databases, based on extensive and standardised data collection, obtaining robust evidence with little expenditure of time and money. We have better emphasised these aspects among the strengths of our study, and added a sentence in the conclusions concerning the potential use of the evidence produced.

The disadvantages are essentially related precisely to the type of data sources, as already discussed among the limitations, i.e. to the lack of clinical data on patients or information on the therapeutic indication and the mode of intake determined by the doctor. This supports the need for further studies, as highlighted in the Conclusions.

“Since fluoroquinolones were among the most commonly used antibiotics and subject of a great deal of attention by regulatory authorities”; add a reference.

Please note that the reference is the number 16, reported after the brackets (EMA. Quinolones and fluoroquinolones Art. 31 PhV - Annex III, https://www.ema.europa.eu/en/documents/referral/quinolone-fluoroquinolone-article-31-referral-annex-iii_en.pdf).

The inspiration of your work must further be highlighted.

We added the following sentence to the Conclusions:

Our data could contribute to fill the gap in knowledge in this field, in the attempt to provide the basis of tailored interventions.

Future work must be included.

We added the following sentence to the Conclusions:

The limitations discussed above and the global impact of the problem, however, call for further studies that can, on the one hand, provide more insight into the areas and determinants of inappropriate antibiotic use and, on the other hand, assess the effective-ness of improvement measures in the medium to long term.

Round 2

Reviewer 2 Report

Accept.